# Hematology Reference Intervals for Holstein Cows in Southern China: A Study of 786 Subjects

**DOI:** 10.3390/vetsci9100565

**Published:** 2022-10-13

**Authors:** Hongbo Chen, Bo Yu, Chenhui Liu, Lei Cheng, Jie Yu, Xiuzhong Hu, Min Xiang

**Affiliations:** 1Laboratory of Genetic Breeding, Reproduction and Precision Livestock Farming & Hubei Provincial Center of Technology Innovation for Domestic Animal Breeding, School of Animal Science and Nutritional Engineering, Wuhan Polytechnic University, Wuhan 430023, China; 2Institute of Animal Science and Veterinary Medicine, Wuhan Academy of Agricultural Sciences, Wuhan 430208, China

**Keywords:** reference interval, hematology, Holstein cows, surveillance, ages, parities, lactation stages

## Abstract

**Simple Summary:**

The American Society of Veterinary Clinical Pathology guidelines recommend that hematology reference intervals (RIs) should be established based on specific animals in specific regions. However, the information on hematology RIs is still lacking for Chinese dairy cows, which account for a total number of more than 3 million. Thus, the aim of this study is to establish hematology RIs for Chinese dairy cows using a large sample size. After the collection of blood samples from 786 Holstein cows, we generated hematology RIs according to overall animals, different ages, different parities and different lactation stages. Our results provide important hematologic information for both clinicians and researchers to approach the health surveillance of Holstein cows in southern China.

**Abstract:**

Hematology RIs help clinicians and researchers determine whether a hematology parameter is abnormal, which plays an important role in animal health surveillance. China is one of the largest dairy producers in the world, with millions of Holstein cows. However, there has been no published data on hematology RIs for dairy cows in China yet. Therefore, the aim of this study is to establish updated and accurate RIs for Holstein cows in southern China. To increase the accuracy of the RIs, we collected blood samples from 786 Holstein cows and analyzed 25 hematologic variables. The RIs for Holstein cows were established using the 95% percentile RIs according to the American Society of Veterinary Clinical Pathology guidelines. The effects of different ages, parities and lactation stages were also checked in this study. The data of 21, 22 and 19 out of 25 hematology parameters were significantly different between different ages, parities and lactation stages, respectively. Furthermore, the hematology RIs of separate subclasses according to different ages, parities and lactation stages were generated. Hematology RIs according to ages and lactation stages, as well as parities and lactation stages, were also assessed. Together, our results confirm that hematology RIs for cows vary by ages, parities and lactation stages. The present study helps to fill the gap in hematology RIs for Holstein cows in southern China, and our data may serve as a very useful tool for monitoring the health and welfare of dairy cattle in China.

## 1. Introduction

Hematology is a routinely used laboratory test to determine animal health and to diagnose and monitor diseases [1,2,3]. Establishing RIs for different hematology parameters is an essential step in hematology tests to interpret whether a parameter from an individual is abnormal. By definition, a reference internal (or interval) is a set of values that includes central 95% (2.5–97.5%) intervals obtained from the population in a defined healthy state [4]. In the human field, a vast amount of hematology tests performed daily worldwide results in accurate and updated RIs of hematology for each specific region [5]. However, the establishment of hematology RIs in the animal field is still far from optimal and it still faces several practical challenges. It has been reported that bovine practitioners and clinical pathologists are still using hematology RIs published in 1966, since these RIs are referenced in veterinary medical trial books [6,7]. Nevertheless, this study showed that many hematology RIs for bovines underwent significant changes in the years 1957 to 2006 [8]. The Clinical Laboratory Standards Institute (CLSI) recommended that RIs should be established using a minimum of 120 healthy individuals [9]. Due to the cost, hematological testing is not regularly performed on farm animals [8]. Thus, there is still a strong demand for establishing updated RIs of hematology based on large data sets in the animal field. 

Holstein is a breed of large dairy cattle originating in northern Holland and Friesland. Due to the high productivity and superior milk quality, it is now one of the most widespread dairy cattle breeds worldwide. China is home for more than 3 million Holstein cows now. According to our knowledge, however, there is currently no published information about hematology RIs for Holstein cows in China. It has been well documented that hematology RIs vary a lot with genotypes, ages, genders, climates and animal management [10,11,12]. So far, most studies about hematology RIs for Holstein cows have been from European and American regions, with obvious differences in animal genotypes, animal management and climates from China [10,12,13]. Directly adopting hematology RIs for Holstein cows from other regions will certainly cause inaccuracies in healthy determination or misjudgment of disease diagnosis in China.

Therefore, the present study was designed to establish reliable and updated hematology RIs for Holstein cows in Southern China. To increase the accuracy of the RIs, we collected blood samples from more than 700 clinically healthy Holstein cows in Hubei, China. The hematology values were checked using a hematology analyzer and the hematology RIs for Holstein cows were determined accordingly. 

## 2. Materials and Methods

### 2.1. Animals

A total of 786 Holstein cows from a large-scale dairy farm (Wuhan, Hubei, P.R. China) were included in this study. The climate corresponds to northern subtropical monsoon climate (latitude 30°34′ N–30°47′ N, longitude 113°53′ E–114°30′ E) with an mean annual temperature of 17.3 °C (−5–39.7 °C), a non-frost period of 266 days per year and an average annual rainfall of 1195.8 mm (rainfall days: 140 days per year). All dairy cows were housed inside deep bedded cubicles with additional loafing areas, which means the cows could feed, drink, congregate and socialize freely. The cows were regularly vaccinated with the foot-and-mouth disease (FMD) vaccine (serotypes of O and A, strains of OHM/02 + AKT-Ⅲ; TECON Biology, Wulumuqi, China) and the inactivated bovine epidemic hemorrhagic fever (BEHF) vaccine (Harbin Weike Biotechnology, Harbin, China) according to the strict regulations of the farm. The cows were fed with the standard TMR diet formulated in accordance with the NRC (2001) (main ingredients: clean water, compound feed, silage corn, corn, cottonseed cake, beet pulp, fodder grasses, etc.) twice a day and milked three times daily (average milk yield: about 30 kg for each dairy cow). In the farm, for cows, the average age of the first insemination and pregnancy were about 14 months and 15 months, respectively; and the average incidents of FMD foot-and-mouth disease and mastitis were about 1.5% and 4%, respectively. Animals were closely monitored daily for their body condition, diet, drinking, body temperature and behavior, and were examined by a clinician prior to scheduled sampling. To make sure that the animals in this study had no hoof and udder health problems, the Dairy Herd Improvement (DHI) Program was carried out regularly. Cows with somatic cell counts (SCCs) < 500,000 cells/ml were actually considered to be uninfected according to the health management program in the farm. To better rule out 90the presence of inflammation, the round 2 test was performed using CMT Mastitis Complete Kit (NEOGEN^®^, Lansing, MI, USA) for the cows with SSCs of 400,000–500,000 cells/mL. 

### 2.2. Blood Sample Collection and Laboratory Analysis

The time interval between vaccination and blood collection for each cow was of at least 30 days. Blood sample collection and hematology tests were performed as described previously [14]. To be more specific, blood samples from the cows were collected via coccygeal vein using a 10 mL disposable blood collection container for animals (SanMen MinSheng Medical Appliances Co., Ltd., Taizhou, China) connected to EDTA tubes (Wuhan Zhiyuan Medical Treatment Technology Co., Ltd., Wuhan, China). All the blood samples were stored at 4 °C and were analyzed within 8 h after collection. A total of 25 hematological parameters including complete blood count (CBC) and white blood cell differential (WBC DIFF) for each blood sample were analyzed using ADVIA^®^ 2120i Hematology System (Siemens Healthcare Diagnostics Inc., Norwood, MA, USA) according to the manufacturer’s instructions. The measurements included white blood cell count (WBC, 10^9^/L), total red blood cell count (RBC, 10^12^/L), hemoglobin concentration (HGB, g/dL), hematocrit (HCT, %), mean corpuscular volume (MCV, fL), mean corpuscular hemoglobin (MCH, pg), mean corpuscular hemoglobin concentration (MCHC, g/dL), cellular hemoglobin concentration mean (CHCM, g/dL), corpuscular hemoglobin (CH, pg), red blood cell distribution width (RDW, %), hemoglobin distribution width (HDW, g/dL), platelet count (PLT, 10^9^/L), mean platelet volume (MPV, fL), total neutrophil count (#NEUT, 10^9^/L), total eosinophils count (#EOS, 10^9^/L), total basophils count (#BASO, 10^9^/L), total lymphocytes count (#LYMPH, 10^9^/L), total monocytes count (#MON, 10^9^/L), total large unstained cells count (#LUC, 10^9^/L), percentage of neutrophils (%NEUT), percentage of eosinophils (%EOS), percentage of basophils (%BASO), percentage of lymphocytes (%LYM), percentage of monocytes (%MON) and percentage of large unstained cells (%LUC).

### 2.3. Statistical Analysis

The cows were grouped according to ages, parities (1 and above) and lactation stages at the sampling time. The age of the tested cows included 2, 3, 4 and 5+ (5 and over 5) years old, with the average age of 3.29 years and an age range of 2–6 years. The parity was defined as first parity (1) and second parity and beyond (2+) with the average parity 1.64 and the range of 1–3. The lactation stage included phase 0 (dry milk period), phase 1 (<100 days), phase 2 (100–200 days) and phase 3 (>200 days).

A normality distribution for each hematological parameter was carried out using univariate progress of SAS online program (https://welcome.oda.SAS.com/, accessed on 15 July 2022) with Kolmogorov–Smirnov, Cramer–von Mises and Anderson–Darling test. Data with nonnormal distribution were transformed by BOX–COX method, and the outliers were removed by ‘mean ± 3 × standard deviation (SD)’ for each parameter. It was carried out independently for each parameter, instead of removing all data of an individual cow after abnormal data were detected, which is why the n-numbers of the analytes in Table 1 are different. The general linear model (GLM) and multiple comparisons with the SNK model were used to analyze the differences of blood parameters among different age/parity/lactation stage groups. Statistical significance was set at *p* < 0.05, and extreme significance was set at *p* < 0.01.

RIs were calculated with Reference Value Advisor (RefValAdv, v.2.1) freeware, a set of Excel macros that compute RIs from spreadsheet data. This freeware tool created analytic reports containing descriptive statistics, histograms, Q/Q plots, information about outliers, calculated 95% RIs with 90% confidence intervals (CIs) with untransformed data, Box Cox transformations or nonparametric methods, based on normality and symmetry tests, data distribution outliers and sample sizes. 

## 3. Results

### 3.1. Establishment of Hematology RIs for Holstein Cows

A total of 786 cow blood samples were available in the database after removal of coagulation and hemolysis samples. After the adoption of exclusion criteria (more than 3 × SD above the mean), the sample number, the mean (± SD) values of the hematology parameters and the hematology RIs for Holstein cows are displayed in Table 1.

### 3.2. Differences of Hematology RIs between Ages, Parities and Lactation Stages

It has been well acknowledged that hematology values are affected by many factors. Therefore, we examined the hematology values of Holstein cows at different ages, parities and lactation stages. 

The results of different age groups revealed that 21 out of 25 hematologic parameters were significantly different (*p* < 0.05) (Table 2). Among them, 18 hematologic parameters showed extremely significant differences (*p* < 0.01). Interestingly, the values of MCV, MCH and CH increased significantly with age. Conversely, a few analytes (MPV, %BASO, %LUC and #BASO) were not significantly different at different ages. 

When cows were sorted by different parities, 22 out of 25 hematologic parameters showed significant differences between primiparous and multiparous cows (Table 3), which was similar to the change law between different ages. The mean values of RBC, HGB, HCT, MCHC, CHCM, RDW, HDW, %NEUT, %MONO, %EOS, %BASO, #NEUT, #MONO and #EOS were significantly higher in primiparous cows, compared to multiparous cows. Consistently, 19 out of 25 analytes were significantly different at different lactation stages (Table 4). Thus, it is necessary to establish hematology RI parameters for Holstein cows into separate subclasses according to different ages, parities and lactation stages.

### 3.3. Differences of Hematology RIs Based on Two Factors

Since we found that most of hematology parameters differed significantly at different ages, parities and lactation stages, we assumed that hematology analytes would also be different based on two factors. This time, we focused on the age and lactation stages, as well as on the parity and lactation stages, because the patterns of hematology parameters based on different ages were quite similar with different parities as described above.

As expected, 17 out of 25 hematologic parameters showed significant differences when cows were sorted by parities and lactation stages (Table 5). The hematology RI parameters are accordingly presented in Table 6 (details on 90% CI lower/upper limits were shown in Appendix A). Likewise, 16 out of 25 hematologic parameters were significantly different when cows were sorted by age and lactation stages, and the hematology RIs were assessed as well (Appendix A). 

## 4. Discussion

Even though the Chinese dairy industry is highly dependent on imports of raw milk products from foreign countries [15,16,17], China is still one of the major dairy producers in the world, with 36.8 million metric tons of milk produced in 2021 (https://www.statista.com/statistics/275794/milk-production-in-china/, accessed on 12 October 2022). Despite the millions of dairy cows in China, there is still no published data about hematology RIs for dairy cows in China. Therefore, we focus on the establishment of hematology RIs for Holstein cows, which represent the majority of dairy cows in China.

Hematology testing is one of the most effective ways to monitor the overall physical conditions of an individual, which is widely used in both human and animal fields. Comparing present hematology results with RIs is the essential step to determine whether a hematology parameter is abnormal. Thus, the accuracy of RIs directly affects the judgment of hematology results, which may change diagnostic decisions. It has been reported that sample size is one of the most important factors for RI determination and a sample size of more than 120 is recommended for establishing RIs [8,18,19]. Despite many studies focusing on establishing hematology RIs for different cattle breeds in different regions, few studies have a relatively large sample size [20,21,22]. Moreover, the sample size turned out to be even smaller when the cattle were sorted by different ages, parities or lactation stages [12,23,24]. Thus, in this study, we included more than 700 Holstein cows for RIs establishment, giving us a chance to still have a relatively large sample size after sorting by ages, parities or lactation stages. 

After outlier removal, we first assessed the hematology RIs using data from all the 786 Holstein cows. In our study, the RI of RBC is 4.81–7.45 *10^12^/L for Holstein cows in southern China. Similar RIs of RBC for cows have been reported in other regions or cattle breeds, namely 4.8–7.8 *10^12^/L for adult cows in France, 4.6–6.9 *10^12^/L for Norwegian Red cows in Norway and 5.1–7.6 *10^12^/L for adult dairy cows in California [8,20,23]. However, the RI of WBC for cows in the present study varies markedly from other studies. The RIs of WBC are 4.4–10.8 *10^9^/L for adult cows in France, 4.7–11.4 *10^9^/L for Norwegian Red cows in Norway and 4.9–12.0 *10^9^/L for adult dairy cows in California [8,20,23]. In contrast, the RI of WBC for Holstein cows in our study is 5.99–20.47 *10^9^/L, which is evidently higher than the RIs of WBC in the studies mentioned above; and the RIs of NEUT and LYMPH are also much higher, compared to the previous studies [8,20,23]. It has been suggested that the total amount of WBC is usually higher in animals under three years old, compared to older ones [23]. Indeed, the cows included in our study were relatively young, with an average age of less than three years. According to the regulations of the farm, the presence of inflammation was ruled out based on monthly SCC records with consideration of the clinical mastitis (CM) history and the round 2 tests of mastitis antigen. Any cows suspected of inflammation were isolated from dairy herds in time for better production management. Therefore, the Holstein cows included in the present study were clinically healthy with no inflammation and the established hematology RIs are reliable. 

It should be emphasized that hematology RIs may be affected by many factors, including seasons [12], production types [23], breeds [24], parities and calving periods [12,25], managements [26] and even different sample medias and lactation stages [12,27], besides age which has been well discussed by many research teams [20,21,23,24]. Therefore, it is possible that the different results between our study and other published studies were due to one or more influencing factors (e.g., breed, management, region, sampling and instrument). As far as we know, there are few investigations about the effects of climatic conditions on hematology RIs establishment, including but not limited to seasons, which need more research in the future. 

As mentioned above, hematology parameters may vary with different ages, parities and lactation stages, and the hematology RIs should be established based on those categories [26]. Our comparisons of hematology parameters into separate subclasses according to different ages showed that the average of WBC increased, but the average of RBC decreased over the ages. Accordingly, the hematology RIs of WBC for Chinese Holstein cows are 5.04–16.66 × 10^12^/L at the age of 2, and 6.38–21.61 × 10^12^/L at the age of 4. Herein, the WBC result is contrary to previous reports in which the total number of WBC decreases with age in cattle (beef and dairy cattle) [23,28]. One explanation for this discrepancy may be the genetic differences of dairy cows from different herds. While the logic is consistent when we assess WBC RIs as a whole (see previous discussion paragraph), that is to say, when all cows are grouped together, implying that one should be cautious when interpreting the laboratory data based on different assessment strategies. 

As for RBC in our study, the hematology RIs of RBC are 5.35–7.95 × 10^9^/L for the cows at age 2, and 4.72–6.95 × 10^9^/L for the cows over 5 years of age. Consistently, it has been suggested that young calves may have higher levels of RBC, compared to adult cows [23,28]. The hematology parameters of different parities showed a similar pattern with those of different ages, since multiparous cows are usually older than primiparous cows. Moreover, the values of MCV, MCH and CH were significantly increased with age in our study. Similarly, it has been reported that the values of MCV, MCH and MCHC are lower in calves compared to adult cows [20,28]. The RIs of WBC increase from 5.94–19.84 *10^12^/L for primiparous cows to 6.10–21.67 *10^12^/L for multiparous cows. Meanwhile, the RIs of RBC decrease slightly from 5.94–7.84 *10^9^/L for primiparous cows to 4.51–7.01 *10^9^/L for multiparous cows. Similarly, it has been reported that the RIs of erythrocytes for primiparous cows are significantly higher than those for multiparous cows in Colombia [29]. When it comes to lactation stages, our study shows that hematology RIs vary a lot at different lactation stages, while other studies generally look at only one stage of lactation, such as the early lactation stage or 30 to 150 days in milk [10,22]. Although there may be overlapping RIs among different study groups, differences may be unavoidable in most cases when directly comparing data sets, especially for the peripheral blood cells in which the gene expression is under highly genetic control [30]. For a precise interpretation of laboratory results, it is better to establish one’s own hematology RIs using local animals under certain specific circumstances.

In the present study, we also assessed hematology RIs based on two factors, namely age and lactation stages, as well as parities and lactation stages. Interestingly, most of the hematology parameters showed significant difference when the cows were sorted by two factors, indicating that it is necessary to establish hematology RIs according to different characteristics of the animals in the future. The limitation of the present study is that all animal samples were derived from a single large-scale dairy farm, which could affect the results. It would be more representative if more samples were collected from several other dairy farms. The fact is that, due to climate reasons, there are few large-scale Holstein farms in southern China. Additionally, the genetic backgrounds, animal management and feeding of Holstein dairy cows also varies among different small farms, with factors that could also affect the establishment of hematology RIs. 

## 5. Conclusions

To conclude, through analysis of hematology data derived from 786 cows, we established hematology RIs for Holstein cows in southern China. We found that most hematology parameters were significantly different when the cows were sorted by different ages, parities and lactation stages, and the hematology RIs were determined accordingly. We also verified the differences of hematology parameters and assessed the hematology RIs based on two factors (age and lactation stage; parity and lactation stage). Our results provide important information of hematologic reference for Holstein cows in southern China.

## Figures and Tables

**Table 1 vetsci-09-00565-t001:** Hematology values and RIs for Holstein cows.

Blood Analytes	N	Mean ± SD	95% RI	90% CI Lower Limit	90% CI Upper Limit
WBC (10^9^/L)	770	10.77 ± 3.56	5.99–20.47	5.54–6.36	19.17–21.70
RBC (10^12^/L)	774	6.20 ± 0.71	4.81–7.54	4.75–4.92	7.39–7.77
HGB (g/dL)	780	104.88 ± 8.51	86.00–120.00	84.00–88.53	118.00–121.97
HCT (%)	777	29.31 ± 2.33	24.50–33.90	24.10–25.00	33.50–34.10
MCV (fL)	776	47.69 ± 4.59	39.74–57.36	39.34–40.10	56.17–58.40
MCH (pg)	776	17.04 ± 1.49	14.50–20.10	14.30–14.60	19.86–20.40
MCHC (g/dL)	775	357.15 ± 10.45	338.00–375.00	336.00–340.00	373.00–377.00
CHCM (g/dL)	781	411.43 ± 9.25	393.00–429.00	390.00–395.00	427.00–431.00
CH (pg)	777	19.52 ± 1.63	16.60–22.90	16.50–16.70	22.60–23.06
RDW (%)	768	20.03 ± 1.48	17.90–23.60	17.80–18.00	23.20–23.90
HDW (g/dL)	777	27.18 ± 1.67	23.90–30.40	23.65–24.10	30.26–30.60
PLT (10^9^/L)	782	618.61 ± 189.02	273.15–980.70	228.45–309.29	947.28–997.55
MPV (fL)	785	13.55 ± 2.94	8.70–19.20	7.76–9.00	18.90–19.60
%NEUT (%)	779	38.18 ± 10.51	18.20–60.10	16.50–19.75	57.80–61.20
%LYMPH (%)	781	48.82 ± 11.26	26.96–72.40	26.08–29.26	70.60–73.70
%MONO (%)	774	5.91 ± 2.05	2.74–10.70	2.60–2.94	10.26–10.90
%EOS (%)	772	4.59 ± 2.62	1.03–11.00	0.90–1.20	10.27–12.30
%BASO (%)	777	1.12 ± 0.31	0.60–1.76	0.60–0.60	1.70–1.80
%LUC (%)	755	0.63 ± 1.11	0.10–4.60	0.10–0.10	4.11–5.01
#NEUT (10^9^/L)	778	3.99 ± 1.23	1.65–6.80	1.43–1.95	6.56–7.31
#LYMPH (10^9^/L)	769	5.35 ± 2.70	2.24–13.01	2.11–2.34	12.21–13.53
#MONO (10^9^/L)	773	0.62 ± 0.22	0.26–1.19	0.24–0.30	1.11–1.23
#EOS (10^9^/L)	773	0.48 ± 0.28	0.10–1.21	0.09–0.12	1.15–1.26
#BASO (10^9^/L)	773	0.12 ± 0.06	0.05–0.31	0.04–0.05	0.27–0.34
#LUC (10^9^/L)	754	0.08 ± 0.14	0.01–0.57	0.01–0.01	0.52–0.61

Note: N = number, SD = standard deviation, RI = reference interval, WBC = white blood cell, RBC = red blood cell, HGB = hemoglobin, HCT = hematocrit, MCV = mean corpuscular volume, MCH = mean corpuscular hemoglobin, MCHC = mean corpuscular hemoglobin concentration, CHCM = cellular hemoglobin concentration mean, CH = corpuscular hemoglobin, RDW = red cell distribution width, HDW = hemoglobin distribution width, PLT = platelets, MPV = mean platelet volume, NEUT = neutrophil, LYMPH = lymphocyte, MONO = monocytes, EOS = eosinophil, BASO = basophilic granulocyte, LUC = unstained large cells, ‘%’ = percentage value, ‘#’ = absolute value (the same below).

**Table 2 vetsci-09-00565-t002:** Hematology values and RIs for Holstein cows at different ages.

Blood Analytes	2 Years (N = 177)		3 Years (N = 278)		4 Years (N = 241)		5+ Years (N = 90)	
	Mean ± SD	95% RI	Mean ± SD	95% RI	Mean ± SD	95% RI	Mean ± SD	95% RI
WBC (10^9^/L)	9.70 ± 2.62 ^A^	5.04–16.66	10.82 ± 3.43 ^B^	6.42–20.40	11.39 ± 3.81 ^B^	6.38–21.61	11.06 ± 4.34 ^B^	4.72–22.71
RBC (10^12^/L)	6.67 ± 0.59 ^C^	5.35–7.95	6.37 ± 0.61 ^B^	4.99–7.58	5.83 ± 0.65 ^A^	4.44–7.05	5.75 ± 0.55 ^A^	4.72–6.95
HGB (g/dL)	105.78 ± 8.14 ^B^	86.75–127.00	106.73 ± 8.15 ^B^	89.78–120.08	102.93 ± 8.78 ^A^	84.05–118.95	102.69 ± 8.12 ^A^	83.75–116.75
HCT (%)	29.18 ± 2.23 ^A^	24.17–34.46	29.78 ± 2.29 ^B^	24.99–33.82	28.98 ± 2.44 ^A^	24.20–34.00	29.02 ± 2.20 ^A^	24.18–33.45
MCV (fL)	44.02 ± 3.11 ^A^	38.94–52.30	47.05 ± 4.01 ^B^	39.49–55.26	49.87 ± 4.29 ^C^	41.70–60.60	50.97 ± 3.92 ^D^	42.50–59.73
MCH (pg)	15.92 ± 1.27 ^A^	13.97–18.70	16.87 ± 1.27 ^B^	14.59–19.50	17.70 ± 1.33 ^C^	15.00–20.41	18.05 ± 1.30 ^D^	15.43–20.68
MCHC (g/dL)	360.89 ± 7.85 ^C^	341.30–373.70	358.49 ± 9.01 ^B^	339.00–377.10	354.43 ± 11.28 ^A^	336.00–375.98	353.21 ± 13.18 ^A^	332.50–366.75
CHCM (g/dL)	415.31 ± 7.73 ^C^	400.00–429.00	412.78 ± 8.56 ^B^	394.93–430.08	408.38 ± 9.91 ^A^	388.05–425.95	407.84 ± 8.29 ^A^	389.50–425.75
CH (pg)	18.22 ± 1.18 ^A^	16.18–21.26	19.34 ± 1.45 ^B^	16.70–22.30	20.27 ± 1.47 ^C^	17.30–23.29	20.66 ± 1.36 ^D^	17.45–23.38
RDW (%)	20.04 ± 1.46 ^ab^	18.13–23.57	20.20 ± 1.52 ^b^	17.89–23.71	19.95 ± 1.52 ^ab^	17.79–24.43	19.71 ± 1.25 ^a^	17.80–23.04
HDW (g/dL)	27.78 ± 1.43 ^C^	24.93–30.87	27.24 ± 1.65 ^B^	23.70–30.50	26.85 ± 1.79 ^A^	23.42–30.40	26.77 ± 1.52 ^A^	23.93–29.53
PLT (10^9^/L)	609.89 ± 169.67 ^A^	315.00–936.23	593.16 ± 177.87 ^A^	306.40–944.40	635.42 ± 203.15 ^AB^	193.65–1014.45	668.38 ± 206.92 ^B^	181.00–1055.88
MPV (fL)	13.02 ± 2.85	9.15-19.60	13.85 ± 3.08	8.68–19.31	13.60 ± 2.85	7.31–18.80	13.53 ± 2.84	8.44–19.54
%NEUT (%)	42.15 ± 10.44 ^C^	20.23–61.29	38.70 ± 9.84 ^B^	19.66–60.59	35.92 ± 10.19 ^A^	15.91–55.75	34.82 ± 10.98 ^A^	11.73–63.30
%LYMPH (%)	44.01 ± 9.74 ^A^	26.30–63.96	47.67 ± 10.83 ^B^	25.57–72.31	51.77 ± 10.84 ^C^	31.10–73.16	53.81 ± 12.23 ^C^	26.48–78.90
%MONO (%)	6.49 ± 2.12 ^B^	2.96–10.83	6.18 ± 2.00 ^B^	3.20-10.70	5.47 ± 1.92 ^A^	2.59–10.34	5.18 ± 1.92 ^A^	2.53–9.87
%EOS (%)	5.02 ± 2.77 ^B^	1.55–12.05	4.92 ± 2.75 ^B^	1.27–12.34	4.18 ± 2.39 ^A^	0.89–9.56	3.79 ± 2.14 ^A^	0.81–9.46
%BASO (%)	1.13 ± 0.34	0.60–1.86	1.15 ± 0.30	0.70–1.80	1.09 ± 0.27	0.60–1.70	1.07 ± 0.31	0.50–1.80
%LUC (%)	0.51 ± 0.86	0.10–3.60	0.65 ± 1.11	0.10–4.60	0.62 ± 1.10	0.10–4.60	0.84 ± 1.49	0.10–5.78
#NEUT (10^9^/L)	4.07 ± 1.34 ^a^	1.18–7.35	4.08 ± 1.19 ^a^	2.14–7.14	3.94 ± 1.20 ^a^	1.63–6.58	3.62 ± 1.12 ^b^	1.21–5.89
#LYMPH (10^9^/L)	4.27 ± 1.73 ^A^	2.17–9.41	5.24 ± 2.56 ^B^	2.25–12.49	6.06 ± 2.96 ^C^	2.30–13.93	5.90 ± 3.26 ^C^	1.55–13.54
#MONO (10^9^/L)	0.62 ± 0.21 ^BC^	0.26–1.18	0.66 ± 0.24 ^C^	0.31–1.23	0.59 ± 0.22 ^B^	0.25–1.21	0.54 ± 0.20 ^A^	0.20–1.09
#EOS (10^9^/L)	0.48 ± 0.28 ^ab^	0.10–1.18	0.51 ± 0.27 ^b^	0.16–1.22	0.47 ± 0.30 ^ab^	0.09–1.30	0.41 ± 0.24 ^a^	0.08–1.09
#BASO (10^9^/L)	0.11 ± 0.04	0.05–0.23	0.13 ± 0.06	0.05–0.31	0.13 ± 0.06	0.05–0.31	0.13 ± 0.08	0.03–0.36
#LUC (10^9^/L)	0.05 ± 0.07 ^A^	0.01–0.34	0.08 ± 0.13 ^AB^	0.01–0.51	0.09 ± 0.17 ^B^	0.01–0.61	0.10 ± 0.19 ^B^	0.01–0.72

Note: Significant differences between groups are indicated by different letters (upper letters: *p* < 0.01, lower letters: *p* < 0.05). More details on 90% CI Lower/Upper limits are shown in Appendix A, ‘%’ = percentage value, ‘#’ = absolute value (the same below).

**Table 3 vetsci-09-00565-t003:** Hematology values and RIs for Holstein cows at different parities.

Blood Analytes	Parity 1 (N = 419)		Parity 2+ (N = 367)	
	Mean ± SD	95% RI	Mean ± SD	95% RI
WBC (10^9^/L)	10.31 ± 3.17 ^B^	5.94–19.84	11.28 ± 3.89 ^A^	6.10–21.67
RBC (10^12^/L)	6.54 ± 0.59 ^B^	5.30–7.84	5.82 ± 0.63 ^A^	4.51–7.01
HGB (g/dL)	106.86 ± 7.75 ^B^	90.00–122.00	102.64 ± 8.79 ^A^	83.00–117.83
HCT (%)	29.68 ± 2.20 ^B^	24.94–34.00	28.90 ± 2.42 ^A^	23.93–33.59
MCV (fL)	45.70 ± 3.93 ^A^	39.30–54.06	49.95 ± 4.24 ^B^	41.61–59.46
MCH (pg)	16.44 ± 1.34 ^A^	14.30–19.30	17.74 ± 1.34 ^B^	15.00–20.59
MCHC (g/dL)	359.52 ± 8.67 ^B^	340.33–375.68	354.47 ± 11.60 ^A^	334.20–374.00
CHCM (g/dL)	413.78 ± 8.44 ^B^	396.43–430.58	408.74 ± 9.41 ^A^	389.00–425.85
CH (pg)	18.83 ± 1.43 ^A^	16.44–21.86	20.31 ± 1.47 ^B^	17.30–23.39
RDW (%)	20.19 ± 1.51 ^B^	18.03–23.68	19.85 ± 1.43 ^A^	17.80–23.50
HDW (g/dL)	27.47 ± 1.55 ^B^	24.46–30.50	26.86 ± 1.75 ^A^	23.45–30.40
PLT (10^9^/L)	594.29 ± 171.15 ^A^	315.00–923.00	646.25 ± 204.22 ^B^	186.18–1018.48
MPV (fL)	13.56 ± 3.04	8.90–19.35	13.54 ± 2.83	8.02–19.08
%NEUT (%)	39.92 ± 10.09 ^B^	19.80–60.12	36.19 ± 10.65 ^A^	15.95–60.30
%LYMPH (%)	46.29 ± 10.28 ^A^	26.78–66.58	51.68 ± 11.65 ^B^	27.19–74.11
%MONO (%)	6.38 ± 2.06 ^B^	3.24–10.70	5.39 ± 1.90 ^A^	2.60–9.90
%EOS (%)	5.05 ± 2.76 ^B^	1.40–12.42	4.04 ± 2.33 ^A^	0.80–9.51
%BASO (%)	1.15 ± 0.32 ^B^	0.60–1.80	1.08 ± 0.28 ^A^	0.60–1.70
%LUC (%)	0.58 ± 0.99	0.10–4.19	0.69 ± 1.23	0.10–5.10
#NEUT (10^9^/L)	4.07 ± 1.26 ^b^	1.70–7.23	3.89 ± 1.19 ^a^	1.61–6.58
#LYMPH (10^9^/L)	4.84 ± 2.31 ^A^	2.24–11.96	5.93 ± 2.99 ^B^	2.22–13.25
#MONO (10^9^/L)	0.65 ± 0.23 ^B^	0.30–1.23	0.58 ± 0.21 ^A^	0.25–1.13
#EOS (10^9^/L)	0.50 ± 0.27 ^b^	0.12–1.21	0.45 ± 0.28 ^a^	0.09–1.22
#BASO (10^9^/L)	0.12 ± 0.06	0.05–0.27	0.13 ± 0.07	0.04–0.34
#LUC (10^9^/L)	0.07 ± 0.12 ^a^	0.01–0.50	0.09 ± 0.17 ^b^	0.01–0.62

Note: Significant differences between groups are indicated by different letters (upper letters: *p* < 0.01, lower letters: *p* < 0.05). More details on 90% CI Lower/Upper limits are shown in Appendix A, ‘%’ = percentage value, ‘#’ = absolute value (the same below).

**Table 4 vetsci-09-00565-t004:** Hematology values and RIs for Holstein cows at different lactation stages.

Blood Analytes	Phase 0 (Dry, N = 40)		Phase 1 (<100, N = 143)		Phase 2 (100–200, N = 293)		Phase 3 (>200, N = 310)	
	Mean ± SD	95% RI	Mean ± SD	95% RI	Mean ± SD	95% RI	Mean ± SD	95% RI
WBC (10^9^/L)	10.19 ± 2.90 ^A^	6.19–18.70	10.00 ± 3.05 ^A^	5.32–19.71	10.31 ± 3.05 ^A^	5.96–18.05	11.61 ± 4.09 ^B^	6.22–22.95
RBC (10^12^/L)	5.64 ± 0.52 ^A^	4.77–7.03	6.20 ± 0.59 ^BC^	4.95–7.21	6.37 ± 0.75 ^C^	4.84–7.91	6.11 ± 0.68 ^B^	4.51–7.36
HGB (g/dL)	103.98 ± 6.86	90.03–115.95	103.58 ± 9.14	82.58–126.43	105.24 ± 8.37	89.00–122.00	105.27 ± 8.51	85.70–118.30
HCT (%)	29.60 ± 2.20	26.00–34.10	28.93 ± 2.40	24.37–33.98	29.22 ± 2.28	24.41–34.15	29.55 ± 2.36	24.40–33.60
MCV (fL)	52.77 ± 4.51 ^C^	41.76–62.16	46.96 ± 3.77 ^A^	40.75–54.30	46.34 ± 4.72 ^A^	39.10–57.24	48.63 ± 4.14 ^B^	40.67–56.93
MCH (pg)	18.53 ± 1.31 ^C^	15.13–21.30	16.78 ± 1.21 ^A^	14.60–19.20	16.66 ± 1.61 ^A^	14.20–20.34	17.33 ± 1.32 ^B^	14.73–20.04
MCHC (g/dL)	351.65 ± 8.59 ^A^	324.25–371.78	357.10 ± 7.68 ^BC^	338.00–373.00	359.81 ± 9.11 ^C^	340.00–378.00	355.43 ± 12.18 ^B^	335.18–375.00
CHCM (g/dL)	403.48 ± 9.24 ^A^	387.03–430.78	410.87 ± 7.94 ^B^	393.53–427.00	413.92 ± 9.04 ^C^	395.33–431.00	410.35 ± 9.21 ^B^	392.25–426.00
CH (pg)	21.13 ± 1.44 ^C^	17.82–24.28	19.22 ± 1.38 ^A^	16.90–21.95	19.07 ± 1.68 ^A^	16.30–22.37	19.88 ± 1.49 ^B^	17.24–22.90
RDW (%)	19.78 ± 1.38 ^B^	17.38–23.00	19.33 ± 0.98 ^A^	17.77–22.20	20.38 ± 1.49 ^C^	18.11–23.69	20.06 ± 1.58 ^BC^	17.80–24.08
HDW (g/dL)	26.78 ± 2.05 ^a^	22.71–32.25	27.08 ± 1.70 ^ab^	23.81–31.40	27.41 ± 1.68 ^b^	23.93–30.58	27.07 ± 1.59 ^ab^	23.97–30.00
PLT (10^9^/L)	693.33 ± 169.61 ^B^	186.88–1016.63	647.55 ± 179.75 ^AB^	324.65–955.50	605.14 ± 186.90 ^A^	272.65–981.05	608.45 ± 194.63 ^A^	190.50–996.50
MPV (fL)	14.99 ± 2.17 ^B^	10.25–19.20	12.94 ± 2.80 ^A^	7.94–19.08	13.62 ± 2.90 ^A^	9.14–19.50	13.59 ± 3.08 ^A^	7.53–19.17
%NEUT (%)	34.39 ± 9.00 ^A^	13.91–51.06	39.77 ± 11.95 ^B^	12.14–61.60	39.59 ± 10.23 ^B^	19.95–60.48	36.60 ± 9.93 ^AB^	17.05–56.98
%LYMPH (%)	52.58 ± 9.07 ^B^	37.11–72.36	46.68 ± 11.89 ^A^	23.69–67.07	47.32 ± 11.00 ^A^	26.76–72.32	50.71 ± 11.08 ^B^	26.98–73.60
%MONO (%)	5.40 ± 1.71 ^A^	2.78–9.91	6.32 ± 2.26 ^B^	2.80–12.25	5.98 ± 1.88 ^AB^	3.13–9.97	5.73 ± 2.12 ^AB^	2.60–10.90
%EOS (%)	5.11 ± 2.94	0.77–12.59	4.25 ± 2.76	0.75–12.13	4.91 ± 2.61	1.40–11.28	4.37 ± 2.49	0.96–10.62
%BASO (%)	1.04 ± 0.38	0.50–2.67	1.11 ± 0.36	0.60–1.89	1.12 ± 0.28	0.60–1.70	1.13 ± 0.29	0.60–1.80
%LUC (%)	0.76 ± 1.19	0.09–4.14	0.73 ± 1.41	1.19–7.34	0.57 ± 1.01	0.10–4.59	0.63 ± 1.03	0.10–4.25
#NEUT (10^9^/L)	3.41 ± 1.08 ^a^	1.46–5.84	3.97 ± 1.42 ^b^	1.19–7.34	3.96 ± 1.15 ^b^	1.72–6.55	4.09 ± 1.20 ^b^	2.03–7.15
#LYMPH (10^9^/L)	5.45 ± 2.11 ^AB^	2.46–11.36	4.61 ± 2.06 ^A^	2.30–10.90	4.94 ± 2.33 ^A^	2.13–11.76	6.06 ± 3.16 ^B^	2.28–15.11
#MONO (10^9^/L)	0.58 ± 0.29	0.23–1.31	0.61 ± 0.22	0.26–1.24	0.61 ± 0.20	0.26–1.05	0.63 ± 0.24	0.29–1.21
#EOS (10^9^/L)	0.53 ± 0.32 ^a^	0.05–1.57	0.42 ± 0.27 ^b^	0.08–1.17	0.49 ± 0.27 ^ab^	0.13–1.20	0.49 ± 0.28 ^ab^	0.13–1.22
#BASO (10^9^/L)	0.10 ± 0.05 ^A^	0.01–0.20	0.11 ± 0.05 ^A^	0.05–0.32	0.12 ± 0.05 ^A^	0.05–0.24	0.14 ± 0.07 ^B^	0.05–0.36
#LUC (10^9^/L)	0.13 ± 0.22 ^a^	0.01–0.89	0.08 ± 0.18 ^ab^	0.01–0.76	0.06 ± 0.11 ^b^	0.01–0.45	0.08 ± 0.14 ^ab^	0.01–0.57

Note: Significant differences between groups are indicated by different letters (upper letters: *p* < 0.01, lower letters: *p* < 0.05). More details on 90% CI Lower/Upper limits are shown in Appendix A, ‘%’ = percentage value, ‘#’ = absolute value (the same below).

**Table 5 vetsci-09-00565-t005:** Hematology values for Holstein cows when sorted by parities and lactation stages.

Blood Analytes	Parity 1 (Mean ± SD)				Parity 2+ (Mean ± SD)			
	Phase 0 (N = 19)	Phase 1 (N = 88)	Phase 2 (N = 178)	Phase 3 (N = 134)	Phase 0 (N = 21)	Phase 1 (N = 55)	Phase 2 (N = 115)	Phase 3 (N = 176)
WBC (10^9^/L)	11.13 ± 3.18 ^B^	9.34 ± 2.14 ^A^	9.89 ± 2.71 ^A^	11.38 ± 3.91 ^B^	9.30 ± 2.34 ^a^	11.06 ± 3.89 ^ab^	10.97 ± 3.43 ^ab^	11.79 ± 4.22 ^b^
RBC (10^12^/L)	5.90 ± 0.53 ^A^	6.42 ± 0.47 ^B^	6.75 ± 0.60 ^C^	6.42 ± 0.55 ^B^	5.40 ± 0.36 ^A^	5.84 ± 0.59 ^B^	5.80 ± 0.57 ^B^	5.88 ± 0.68 ^B^
HGB (g/dL)	105.95 ± 6.65	104.87 ± 7.56	107.06 ± 8.14	108.05 ± 7.28	102.19 ± 6.69	101.53 ± 10.96	102.44 ± 7.97	103.18 ± 8.79
HCT (%)	30.12 ± 2.02	29.15 ± 2.05	29.49 ± 2.31	30.20 ± 2.06	29.12 ± 2.29	28.59 ± 2.85	28.79 ± 2.17	29.05 ± 2.45
MCV (fL)	51.26 ± 3.94 ^D^	45.49 ± 3.30 ^B^	43.98 ± 3.39 ^A^	47.32 ± 3.59 ^C^	54.13 ± 4.64 ^B^	49.35 ± 3.25 ^A^	49.95 ± 4.16 ^A^	49.63 ± 4.26 ^A^
MCH (pg)	18.02 ± 1.07 ^C^	16.31 ± 1.06 ^A^	15.95 ± 1.33 ^A^	16.94 ± 1.22 ^B^	18.99 ± 1.35 ^B^	17.55 ± 1.03 ^A^	17.75 ± 1.37 ^A^	17.63 ± 1.32 ^A^
MCHC (g/dL)	352.05 ± 8.29 ^A^	358.11 ± 7.09 ^B^	362.47 ± 9.04 ^C^	357.65 ± 7.81 ^B^	351.29 ± 9.03	355.46 ± 8.35	355.74 ± 7.62	353.73 ± 14.45
CHCM (g/dL)	407.21 ± 9.05 ^A^	411.69 ± 8.04 ^B^	416.72 ± 7.69 ^C^	412.17 ± 8.27 ^B^	400.10 ± 8.21 ^A^	409.53 ± 7.65 ^B^	409.60 ± 9.28 ^B^	408.98 ± 9.66 ^B^
CH (pg)	20.72 ± 1.28 ^C^	18.67 ± 1.21 ^A^	18.25 ± 1.28 ^B^	19.43 ± 1.32 ^A^	21.50 ± 1.50 ^B^	20.10 ± 1.18 ^A^	20.33 ± 1.42 ^A^	20.22 ± 1.52 ^A^
RDW (%)	19.72 ± 1.57 ^AB^	19.20 ± 0.90 ^A^	20.85 ± 1.46 ^C^	20.05 ± 1.48 ^B^	19.85 ± 1.20	19.55 ± 1.07	19.65 ± 1.21	20.06 ± 1.65
HDW (g/dL)	27.12 ± 1.97	27.18 ± 1.64	27.88 ± 1.43	27.17 ± 1.48	26.48 ± 2.13	26.91 ± 1.80	26.70 ± 1.78	27.00 ± 1.66
PLT (10^9^/L)	700.95 ± 128.12 ^B^	643.11 ± 174.83 ^AB^	573.85 ± 167.17 ^A^	574.31 ± 168.84 ^A^	686.43 ± 203.02	654.69 ± 188.85	653.31 ± 205.34	634.25 ± 208.83
MPV (fL)	14.80 ± 2.09 ^b^	12.77 ± 2.85 ^a^	13.63 ± 2.97 ^ab^	13.82 ± 3.28 ^ab^	15.15 ± 2.27 ^b^	13.22 ± 2.71 ^a^	13.59 ± 2.80 ^a^	13.42 ± 2.91 ^a^
%NEUT (%)	33.85 ± 10.39 ^A^	41.73 ± 11.62 ^B^	41.06 ± 8.99 ^B^	38.08 ± 9.82 ^B^	34.90 ± 7.69	36.62 ± 11.89	37.32 ± 11.59	35.47 ± 9.88
%LYMPH (%)	53.02 ± 10.88 ^B^	44.12 ± 10.03 ^A^	44.92 ± 9.52 ^A^	48.57 ± 10.59 ^A^	52.19 ± 7.31	50.74 ± 13.47	51.04 ± 12.10	52.32 ± 11.19
%MONO (%)	5.91 ± 1.80	6.66 ± 2.41	6.43 ± 1.87	6.19 ± 2.09	4.89 ± 1.49	5.78 ± 1.89	5.31 ± 1.68	5.37 ± 2.07
%EOS (%)	4.87 ± 3.46	4.68 ± 3.07	5.35 ± 2.67	4.93 ± 2.54	5.33 ± 2.41 ^b^	3.53 ± 1.97 ^a^	4.22 ± 2.34 ^a^	3.94 ± 2.37 ^a^
%BASO (%)	1.07 ± 0.47	1.13 ± 0.41	1.15 ± 0.28	1.16 ± 0.28	1.01 ± 0.28	1.08 ± 0.25	1.06 ± 0.28	1.11 ± 0.29
%LUC (%)	0.88 ± 1.33	0.59 ± 1.08	0.49 ± 0.79	0.65 ± 1.10	0.66 ± 1.08	0.95 ± 1.81	0.69 ± 1.28	0.61 ± 0.96
#NEUT (10^9^/L)	3.64 ± 1.18	4.06 ± 1.55	3.99 ± 1.05	4.24 ± 1.29	3.19 ± 0.95 ^a^	3.84 ± 1.19 ^b^	3.91 ± 1.30 ^b^	3.97 ± 1.12 ^b^
#LYMPH (10^9^/L)	6.01 ± 2.41 ^B^	4.00 ± 1.12 ^A^	4.51 ± 1.97 ^A^	5.66 ± 2.93 ^B^	4.92 ± 1.66	5.60 ± 2.75	5.60 ± 2.68	6.36 ± 3.30
#MONO (10^9^/L)	0.67 ± 0.33	0.62 ± 0.22	0.64 ± 0.21	0.69 ± 0.26	0.51 ± 0.23	0.61 ± 0.24	0.56 ± 0.17	0.59 ± 0.21
#EOS (10^9^/L)	0.51 ± 0.34	0.44 ± 0.30	0.52 ± 0.26	0.52 ± 0.25	0.55 ± 0.30	0.39 ± 0.22	0.45 ± 0.27	0.47 ± 0.31
#BASO (10^9^/L)	0.11 ± 0.05 ^A^	0.10 ± 0.04 ^A^	0.12 ± 0.05 ^AB^	0.14 ± 0.07 ^B^	0.10 ± 0.05 ^a^	0.13 ± 0.07 ^b^	0.12 ± 0.06 ^ab^	0.14 ± 0.07 ^b^
#LUC (10^9^/L)	0.16 ± 0.26 ^B^	0.04 ± 0.07 ^A^	0.05 ± 0.08 ^A^	0.09 ± 0.15	0.10 ± 0.18	0.14 ± 0.27	0.08 ± 0.14	0.08 ± 0.14

Significant differences between groups are indicated by different letters (upper letters: *p* < 0.01, lower letters: *p* < 0.05), ‘%’ = percentage value, ‘#’ = absolute value (the same below).

**Table 6 vetsci-09-00565-t006:** Hematology RIs for Holstein cows when sorted by parities and lactation stages.

Blood Analytes	Parity 1 (95% RI)				Parity 2+ (95% RI)			
	Phase 0 (N = 19)	Phase 1 (N = 88)	Phase 2 (N = 178)	Phase 3 (N = 134)	Phase 0 (N = 21)	Phase 1 (N = 55)	Phase 2 (N = 115)	Phase 3 (N = 176)
WBC (10^9^/L)	6.61–20.84	4.77–14.02	5.63–17.41	6.53–23.41	5.73–15.74	5.94–23.06	5.85–18.22	6.15–22.52
RBC (10^12^/L)	4.73–7.04	5.35–7.55	5.35–8.26	5.24–7.74	4.56–6.11	4.77–6.96	4.76–6.98	4.38–7.16
HGB (g/dL)	92.28–122.22	88.00–126.80	88.85–127.58	91.98–119.68	88.75–117.45	76.20–124.60	88.60–119.10	82.40–117.60
HCT (%)	25.55–34.35	24.57–33.78	24.19–34.66	25.23–33.94	25.16–35.42	23.08–34.80	24.93–33.59	23.59–33.33
MCV (fL)	43.25–52.05	40.50–53.55	38.73–52.69	40.30–55.00	43.02–63.15	42.13–54.40	40.73–59.32	41.64–58.98
MCH (pg)	14.85–19.78	14.50–18.87	13.94–18.81	14.60–19.47	16.06–21.88	15.37–19.46	14.79–20.67	15.00–20.10
MCHC (g/dL)	336.04-373.40	343.20–373.00	340.35–382.95	339.35–371.65	331.04–370.96	331.75–372.88	336.85–373.60	324.20–377.20
CHCM (g/dL)	389.25–428.37	393.20–427.80	400.45–431.00	396.35–431.00	380.95–416.21	392.45–424.65	389.00–431.00	386.13–425.58
CH (pg)	18.07–23.57	16.64–21.55	15.99–21.30	16.74–22.07	17.93–24.42	17.31–22.39	17.06–23.43	17.30–23.19
RDW (%)	17.18–24.25	17.50–22.10	18.43–24.07	17.79–23.81	17.41–22.58	17.79–22.45	17.48–22.30	17.80–24.91
HDW (g/dL)	24.03–33.66	23.62–31.23	25.04–30.60	23.83–29.70	21.90–31.06	23.94–31.40	23.08–30.42	24.00–30.23
PLT (10^9^/L)	366.64–938.36	327.00–947.60	308.40–917.80	303.10–912.85	277.09–1158.73	315.50–1085.63	186.40–1020.80	175.25–1059.58
MPV (fL)	10.64–19.77	8.44–19.09	9.30–19.60	7.41–19.46	10.51–20.90	6.80–19.42	8.60–19.26	7.60–19.02
%NEUT (%)	12.47–57.37	14.30–61.84	23.05–59.11	17.46–59.84	17.72–50.87	8.72–63.42	15.14–65.22	16.38–56.32
%LYMPH (%)	34.66–85.60	25.04–65.46	26.79–63.05	25.85–73.34	37.39–68.94	18.62–83.46	25.46–74.63	28.27–73.90
%MONO (%)	2.34–10.13	2.75–12.37	3.34–10.32	3.30–10.90	2.91–9.51	2.32–11.78	2.96–8.71	2.28–10.74
%EOS (%)	1.10–10.20	1.08–12.90	1.64–12.67	1.41–11.88	1.48–12.55	0.70–7.56	1.10–10.19	0.76–9.65
%BASO (%)	0.52–2.73	0.52–2.16	0.70–1.70	0.70–1.77	0.51–1.75	0.64–1.76	0.50–1.70	0.60–1.80
%LUC (%)	0.09–6.95	0.10–5.18	0.10–3.87	0.10–4.37	0.09–8.83	0.10–7.16	0.10–5.26	0.10–4.15
#NEUT (10^9^/L)	1.29–6.38	0.90–7.81	1.97–6.08	2.25–7.59	1.94–6.46	1.49–6.46	1.06–7.99	1.81–6.58
#LYMPH (10^9^/L)	2.38–12.84	2.25–6.89	2.15–10.06	2.26–15.46	2.33–9.53	2.24–14.14	1.81–11.95	2.27–14.76
#MONO (10^9^/L)	0.26–1.74	0.26–1.23	0.30–1.12	0.33–1.31	0.21–1.34	0.27–1.26	0.24–0.90	0.24–1.17
#EOS (10^9^/L)	0.03–1.42	0.08–1.20	0.15–1.20	0.17–1.18	0.19–1.59	0.07–1.06	0.09–1.21	0.09–1.29
#BASO (10^9^/L)	0.04–0.25	0.04–0.20	0.05–0.24	0.05–0.37	0.04–0.24	0.04–0.34	0.04–0.27	0.05–0.36
#LUC (10^9^/L)	0.01–2.16	0.01–0.30	0.01–0.37	0.01–0.54	0.01–0.53	0.01–1.10	0.01–0.60	0.01–0.60

Note: More details on 90% CI Lower/Upper limits are shown in Appendix A, ‘%’ = percentage value, ‘#’ = absolute value (the same below).

## Data Availability

The data that support the findings of this study are available from the corresponding author upon reasonable request.

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
