# Peer review of "Hematology Reference Intervals for Holstein Cows in Southern China: A Study of 786 Subjects"

_vetsci, 2022, doi:10.3390/vetsci9100565_

Round 1
Reviewer 1 Report
The manuscript is well written and the data are important for the medical management of cows in China, but important information are still missing and much more discussion is needed on the causes of the variations.
Line 43-51: The first paragraph could be deleted as it contains only basic general information. You could start with a shorter paragraph like this: "Hematology is a routinely used laboratory test to determine animal health and is also used to diagnose and monitor diseases.
Line 64: The sentence should be rewritten like “Because of the cost, hematological testing are not regularly performed on farm animals.”
Line 70: “widespread dairy cattle breed”
Line 85-90: More information is needed:
-Housing and climatic conditions.
- Please add the vaccines and the period between vaccination and blood collection
- How was the hoof and udder health?
- Please include any information on the TMR (ingredients, energy content, etc.)
- What was the average milk yield in each group?
- Average age of the first insemination/pregnancy
Line 114: Please add the n-number for each group and the mean age and range etc.
Line 115: Was only a certain group of animals selected? Because the division into one and 2+ parities and also the age division give the impression that the animals in this farm do not get very old. Is this normal in China? Please discuss the reasons.
Line 121: Which statistical test were used for the evaluation of normal/ nonnormal distribution?
Line 123: Why do you use this model, because it has a high risk of false positive results?
Line 137-141: This paragraph can be deleted as the data are shown in the table and does not need to be repeated in the text.
Table 1: Why are the n-numbers of the analytes so different? You have no clear exclusion criteria listed in the material and method sections (Please add it). And you show here more data for the differential blood count than for the WBC itself!
Line :161-165: Should be deleted.
Table 5 and 6: In some groups the number of tested samples is too low for the calculation of reference intervals based on the guidelines of the American Society of Veterinary Clinical Pathology.
Discussion: Please discuss more possible reasons for the hematologic differences between the different groups studied!
Line 2019-227: But in some groups the sample size was also under 120 in the present study!
Line 228-245: You only discussed age as a potential influencing factor and not local factors such as climatic conditions, so the statement in the last sentence is not discussed or proved.
Line 249: This statement is in contrast to the statement in your previous paragraph where you indicated that cows under 3 years of age have a higher WBC than older cows.
Reviewer 2 Report
The manuscript entitled "Hematology Reference Intervals for Holstein Cows in Southern 2 China: A Study of 786 Subjects" sounds interesting according to my opinion, even though some issues are to be better argued. According to the results, WBC seems to be the hematological parameter which differs more from the other reported results in scientific works done in France, Norway and California. This result should be better estabilished and other parameters should be evaluated to be sure that a bias is not present. Considering that elevation of WBC count is one of the main finding used to suspect an ongoing inflammatory process, the study should rule out the presence of inflammation through other tests. The Authors stated that that the Holstein cows included in the study were clinically healthy with no inflammation and the RIs established are reliable, but this statement is not supported by objective tests other than a phisycal examination. Total plasma proteins and Albumin:Globulin ratio in association with acute phase proteins should be used to evaluate underlying inflammatory processes. Results showed that the values of MCV, MCH and CH were significantly increased with the increament of age: please try to argue this finding. The animal samples derived from a single large-scale dairy farm and this could affect the rersults of the study.
Round 2
Reviewer 1 Report
Thank you very much for the comprehensive implementation of my comments and the intensive revision of the manuscript. In my opinion, no further changes are necessary in the manuscript.
Author Response
Many thanks for your concern! We are pleased that you are satisfied with our responses!
Reviewer 2 Report
The manuscript has been revised according to the suggestions. Please some grammatical/language inaccurancies should be corrected.
Author Response
Thank you very much for you concern ! And, the language has been rewritten or further edited by three experts. Hopefully, it reads better now.
kindest regards,
Hongbo Chen & Lei Cheng